# D³PG: DEEP DIFFERENTIABLE DETERMINISTIC POLICY GRADIENTS

## ABSTRACT

Over the last decade, two competing control strategies have emerged for solving complex control tasks with high efficacy. Model-based control algorithms, such as model-predictive control (MPC) and trajectory optimization, peer into the gradients of underlying system dynamics in order to solve control tasks with high sample efficiency. However, like all gradient-based numerical optimization methods, model-based control methods are sensitive to initialization and are prone to becoming trapped in local minima. Deep reinforcement learning (DRL), on the other hand, can somewhat alleviate these issues by exploring the solution space through sampling — at the expense of computational cost. In this paper, we present a hybrid method that combines the best aspects of gradient-based methods and DRL. We base our algorithm on the deep deterministic policy gradients (DDPG) algorithm and propose a simple modification that uses true gradients from a differentiable physical simulator to increase the convergence rate of both the actor and the critic. We demonstrate our algorithm on seven 2D robot control tasks, with the most complex one being a differentiable half cheetah with hard contact constraints. Empirical results show that our method boosts the performance of DDPG without sacrificing its robustness to local minima.

## 1 INTRODUCTION

In recent years, deep reinforcement learning (DRL) has emerged as a flexible and robust means of teaching simulated robots to complete complex tasks, from manipulation (Kumar et al., 2016) and locomotion (Haarnoja et al., 2018b), to navigating complex terrain (Peng et al., 2016). Compared with more direct optimization methods such as gradient descent or second-order optimization, DRL naturally incorporates exploration into its planning, allowing it to learn generalizable policies and robust state value estimations across simulated environments. Perhaps the most salient reason for DRL's surge in popularity is its ability to operate on black-box simulators where the underlying dynamics model is not available. DRL's model-free, Monte-Carlo-style methods have made it applicable to a wide range of physical (and non-physical) simulation environments, including those where a smooth, well-behaved dynamical model does not exist. This comes at two striking costs. First, such sampling procedures may be inefficient, requiring a large number of samples for adequate learning. Second, in order to be generally applicable to any model-free environment, underlying dynamical gradients are not used, even if they are available. In other words, valuable information that could greatly aid control tasks is not taken advantage of in these schemes.

When an accurate model of robot dynamics is given, model-based methods such as model-predictive control (MPC) or trajectory optimization have historically been employed. These methods can solve tasks with higher sample efficiency than model-free DRL algorithms. Models provide access to ground-truth, analytical gradients of robot physics without the need for sample-based estimation. However, such methods don't incorporate exploration or learning into their procedures, and are especially prone to becoming trapped in poor local minima.

While there has been a recent surge in fast and accurate differentiable simulators not previously available, most applications for control have relied on established local methods such as MPC (de Avila Belbute-Peres et al., 2018), gradient descent (Degrave et al., 2019), or trajectory

optimization (Hu et al., 2019) to solve control tasks. An ideal algorithm would exploit the efficiency of model-based methods while maintaining DRL's relative robustness to poor local minima.

In this paper, we propose an actor-critic algorithm that leverages differentiable simulation and combines the benefits of model-based methods and DRL. We build our method upon standard actor-critic DRL algorithms and use true model gradients in order to improve the efficacy of learned critic models. Our main insights are twofold: First, gradients of critics play an important role in certain DRL algorithms, but optimization of these critics' gradients has not been explored by previous work. Second, the emergence of differentiable simulators enables computation of advantage estimation (AE) gradients with little additional computational overhead. Based on these observations, we present an algorithm that uses AE gradients in order to co-learn critic value and gradient estimation, demonstrably improving convergence of both actor and critic.

In this paper, we contribute the following: *1)* An efficient hybrid actor-critic method which builds upon deep deterministic policy gradients (DDPG, (Lillicrap et al., 2015)), using gradient information in order to improve convergence in a simple way. *2)* A principled mathematical framework for fitting critic gradients, providing a roadmap for applying our method to any deterministic policy gradient method, and *3)* Demonstrations of our algorithm on seven control tasks, ranging from contact-free classic control problems to complex tasks with accurate, hard contact, such as the HalfCheetah, along with comparisons to both model-based control and DRL baselines.

## 2 RELATED WORK

Model-based control methods peer into the underlying dynamics of a system in order to optimize control parameters. Such approaches have especially grown in popularity with the rise of differentiable simulators, whose application now spans rigid body simulation (de Avila Belbute-Peres et al., 2018; Degrave et al., 2019), soft body simulation (Hu et al., 2019), and fluid dynamics (Schenck & Fox, 2018; Li et al., 2019). Such simulators allow for backward propagating through an entire simulation, and even the simplest optimization algorithms (gradient descent) can be employed to a great effect. By treating robotic control tasks as a nonlinear numerical optimization problem, two more sophisticated strategies, trajectory optimization (Posa et al., 2014; Manchester & Kuindersma, 2017; Winkler et al., 2018; Marchese et al., 2016; Bern et al., 2019) and model-predictive control (MPC) (Todorov & Li, 2005; Tassa et al., 2012; 2014; Lenz et al., 2015), employ established numerical optimization techniques to efficiently solve complex, even high-dimensional control tasks. But, these methods are are prone to becoming trapped in poor local minima, especially in the presence of complex dynamics or terrain.

Separate from model-based control methods, which assume the model of dynamics is known, model-based learning methods (Gu et al., 2016; Yamaguchi & Atkeson, 2016) attempt to learn a prior *unknown* model so as to directly optimize policies. In this paper, we do not focus on model-based learning, but rather the complementary problem, where we assume an accurate model of dynamics is given by a differentiable simulator and we use it to efficiently explore optimal control parameters.

In contrast to the above local methods, stochastic global optimization methods, such as evolutionary algorithms (*e.g.*, (Hansen et al., 2003)) or simulated annealing, explore the solution space with a large number of samples in order to find global optima. Such methods, however, can be quite slow, due to the large number of samples they require. RL, while technically local, naturally incorporates sampling, exploration, and value function learning in a similar way that greatly mitigates optimization's propensity to becoming trapped in poor local minima. Our algorithm similarly incorporates exploration to mitigate the likelihood of poor local minima, while using gradient information to do so with high sample efficiency.

## 3 METHOD

### 3.1 PRELIMINARIES

**Markov-Decision Problems (MDP)**  In this paper, we consider the rigid body simulation as a discrete-time Markov Decision Process (MDP) defined by $(\mathcal{S}, \mathcal{A}, \Gamma, \mathcal{R})$, where $\mathcal{S} \subseteq \mathbb{R}^n$ is the state space, $\mathcal{A} \subseteq \mathbb{R}^m$ is the action space, $\Gamma : \mathcal{S} \times \mathcal{A} \to \mathcal{S}$ is the deterministic dynamic transition function,

and $\mathcal{R} : \mathcal{S} \times \mathcal{A} \times \mathcal{S} \to \mathbb{R}$ is the reward function. The optimal control policy $\pi : \mathcal{S} \to \mathcal{A}$ maximizes the $\gamma$-discounted sum of reward $\eta(\pi)$ over a finite time horizon $T$, which is defined as:

$$\eta(\pi) = \mathbb{E}_{\rho_0, \pi}[\sum_{t=0}^{T-1} \gamma^t r_t].$$

Here $\rho_0$ specifies the distribution of the initial state $s_0$ and $r_t = r(s_t, a_t, s_{t+1})$ represents the reward collected at time step $t$. We represent the policy function $\pi$ as a neural network parameterized by $\theta$ and assume $\pi_\theta$ is a deterministic policy that takes as input the current state $s_t$ and predicts the next action $a_t = \pi_\theta(s_t)$.

The action-value function $Q^\pi(s_t, a_t)$, which describes the expected return given the action $a_t$ and state $s_t$ at step $t$, is defined as

$$Q^\pi(s_t, a_t) = \mathbb{E}_{s_{t+1}, a_{t+1}, \cdots}[\sum_{l=t}^{T-1} \gamma^{l-t} r_l(s_l, a_l, s_{l+1})], \tag{1}$$

in which $a_l \sim \pi(s_l)$ and $s_l \sim \Gamma(s_{l-1}, a_{l-1})$. The expectation can be removed in our case since both $\Gamma$ and $\pi$ are deterministic. This function is approximated by the critic network $Q_\phi(s, a)$ parameterized with $\phi$.

**Differentiable simulators** In this paper, we focus on differentiable and deterministic rigid body simulators with fully observable states. At each time step $t$, the state $s_t$ fully characterizes the current status of the robot. The simulator takes current state $s_t$ and an action $a_t$ as input and returns the next state $s_{t+1}$ by solving the governing physical equations. Additionally, we assume a differentiable simulator also returns a Jacobian matrix $\nabla_{s_t, a_t} s_{t+1}$. For all the differentiable rigid body simulators in this paper, the time and space complexities of computing the Jacobian matrix are the same as computing $s_{t+1}$ in one forward step.

## 3.2 ALGORITHM

Our hybrid algorithm takes inspiration from a subclass of modern actor-critic algorithms which seek to improve actor policies *via* critic gradients (such as DDPG). These algorithms iteratively improve the model of the critic, or estimated future return, using recorded data. Such algorithms implicitly assume that if the critic is well-fit to reality, then the gradients must also be well-fit. If we want to prioritize accurate critic gradients and better bootstrap our actor learning, it is best to ensure this directly. Specifically, we augment DDPG's critic gradient fitting step with a regularization term which prioritizes fitting accurate gradients in addition to accurate returns. Our choice of working in the domain of deterministic policy gradients is deliberate; as will be seen, this framework allows us to use our differentiable simulation engine to back propagate through the state-action pairs of our MDP, generating ground-truth reward gradients to fit.

Algorithm 1 summarizes our method with our modifications to the original DDPG highlighted in color. Like actor-critic methods, our algorithm populates a replay buffer of state-action pairs from which to learn. At the same time, it records the dynamical Jacobian $\nabla_{s_t, a_t} s_{t+1}$ corresponding to those state-action pairs, without degrading its asymptotic time or space complexity. Like the state-action pairs, these Jacobians need only be computed during rollouts; they can be re-used during the embedded supervised training steps to efficiently improve the actor and critic networks.

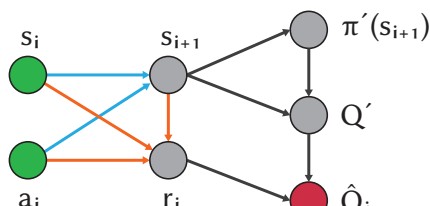

Figure 1: Computation graph for $\hat{Q}_i$. Input and output nodes are colored as green and red. Gradients of blue and orange arrows are computed from a differentiable simulator and the reward definition respectively.

With additional Jacobian information about the dynamics at hand, we compute the gradients of the target network $\hat{Q}_i$. Since $\hat{Q}_i$ defines the target value that the critic $Q_\phi$ attempts to fit to, $\nabla \hat{Q}_i$ provides a good estimation to the desired $\nabla Q$ as well. This motivates us to use it as a regularizer in the critic loss (Line 10 in Algorithm 1) with two additional weights $w_1$ and $w_2$ corresponding to the weights on the two partial

derivatives. The definition of $\nabla_{s_i, a_i} \hat{Q}_i$ is provided by the following equation (For brevity, we use the network names to refer to their output values in the equation):

$$\nabla_{s_i, a_i} \hat{Q}_i = \nabla_{s_i, a_i} r_i + \nabla_{s_{i+1}} r_i \cdot \nabla_{s_i, a_i} s_{i+1} + \gamma(\nabla_{s_{i+1}} Q' + \nabla_{\pi'} Q' \cdot \nabla_{s_{i+1}} \pi') \nabla_{s_i, a_i} s_{i+1}, \quad (2)$$

which can be evaluated in constant time (Figure 1). It is important to note that this computation is only possible when a differentiable simulator is available due to the need for $\nabla_{s_i, a_i} s_{i+1}$.

**Time and space complexity**  Our modifications require extra storage space for two Jacobian matrices of the size $dim(s) \times dim(s)$ and $dim(s) \times dim(a)$ associated with each $s_i$ in the replay buffer, which is tolerable as long as the dimensions of $s$ and $a$ are small constants. This is the case, for example, in rigid robot control tasks. In terms of the time cost, computing $\nabla \hat{Q}_i$ makes efficient use of the cached data. As a result, its computation requires only a handful of matrix multiplications and back-propagation through $Q'$ and $\pi'$.

---

**Algorithm 1** Hybrid DDPG algorithm

---

**Require:** Differentiable simulator $\Gamma$, actor network $\mu_\theta$, critic network $Q_\phi$, weighting factors $w_1$, $w_2$, damping constant $\tau \ll 1$

1: Initialize two target networks $\pi' = \pi_\theta$ and $Q' = Q_\phi$
2: Initialize a replay buffer $\mathcal{B}$
3: Get the initial state $s_0$
4: **while** True **do**
5:     Generate a random process $\mathcal{N}$
6:     Simulate the agent using $a_t = \pi(s_t) + \mathcal{N}_t$ at each time step $t$ until termination
7:     Store all $(s_t, a_t, r_t, s_{t+1}, \nabla_{s_t} s_{t+1}, \nabla_{a_t} s_{t+1})$ to $\mathcal{B}$
8:     Sample a random minibatch of $n$ transitions $(s_i, a_i, r_i, s_{i+1}, \nabla_{s_i} s_{i+1}, \nabla_{a_i} s_{i+1})$ from $\mathcal{B}$
9:     Set $\hat{Q}_i = r_i + \gamma Q'(s_{i+1}, \pi'(s_{i+1}))$.
10:     Update the critic parameter $\phi$ by minimizing the loss:

$$L = \frac{1}{n} \sum_i \left\| Q_\phi(s_i, a_i) - \hat{Q}_i \right\|^2 + w_1 \left\| \nabla_{a_i} Q_\phi - \nabla_{a_i} \hat{Q}_i \right\|^2 + w_2 \left\| \nabla_{s_i} Q_\phi - \nabla_{s_i} \hat{Q}_i \right\|^2$$

11:     Update the actor parameter $\theta$ using the following gradient:

$$\nabla_\theta \eta \approx \frac{1}{n} \sum_i \nabla_a Q_\phi(s_i, a)|_{a=\pi_\theta(s_i)} \nabla_\theta \pi(s_i)$$

12:     Update target networks:

$$\pi' = \tau \pi_\theta + (1 - \tau)\pi'$$
$$Q' = \tau Q_\phi + (1 - \tau)Q'$$

13: **end while**

---

### 3.3  A Motivating Example

We present a simple, analytical example with easily understandable dynamics in a low-dimensional space in order to illustrate the insights behind our algorithm. Consider a one dimensional kinematic mass point whose state $s_t$ and action $a_t$ are two scalars describing its position and velocity respectively. We define $s_0 = -0.5$ and $a_t \in [-1, 1]$. The new state is computed by $s_{t+1} = s_t + a_t \Delta t$ where $\Delta t = 0.01$ is a constant. The goal is to move the mass point to the origin $s = 0$ as quickly as possible. The simulation is terminated when $|s_t| < 0.01$. The reward $r(s, a, s') = (s^2 + 1)^{-1} - 1$ encourages the mass point to converge to the origin in minimal number of steps.

To see how the regularizer affects the training of the critic network, we compare results from four sets of $(w_1, w_2)$ parameters: $(0, 0)$, corresponding to the original DDPG algorithm; $(0, 1)$, which only regularizes $\nabla_{s_i} \hat{Q}_i$; $(1, 0)$, which only regularizes $\nabla_{a_i} \hat{Q}_i$; and $(1, 1)$, which regularizes both $\nabla_{s_i} \hat{Q}_i$ and $\nabla_{a_i} \hat{Q}_i$. We plot the surfaces of intermediate critic network $Q$, their ground truth, and the return-timestep curves in Figure 2. In particular, the surfaces of ground truth $Q$ for the optimal controller $\pi$ are discounted returns directly measured from the forward simulation.

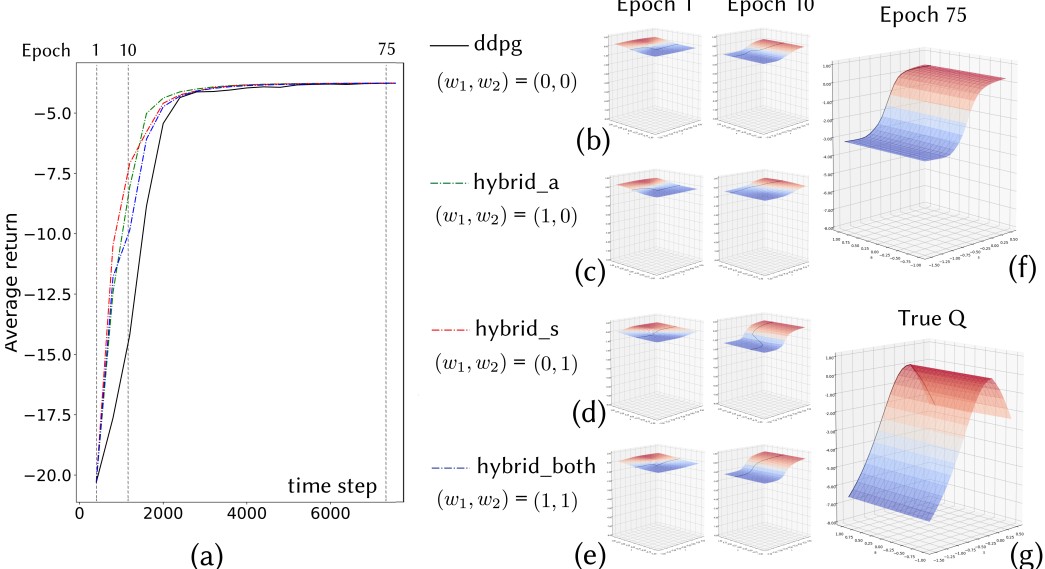

Figure 2: Visualization of critic surfaces: (a) return-timestep curves; (b-e) intermediate critic surfaces at epoch 1 and 10 for original DDPG, only $\nabla_{s_i}\hat{Q}_i$ regularization, only $\nabla_{a_i}\hat{Q}_i$ regularization, and both regularization respectively; (f and g) converged critic surfaces and ground truth of $Q$ at epoch 75. The black curves on $Q$ surfaces demonstrate $\pi_\theta(s)$ at corresponding epochs.

Given these surfaces, the critic update steps in DDPG and the benefits of the regularizer can be explained visually: At each iteration, DDPG updates the critic network to fit the $Q$ surface with points $(s_i, a_i, \hat{Q}_i(s_i, a_i))$, which are sampled from the replay buffer. Additionally, our hybrid method adds the regularizers to also fit the tangent directions of $Q$ surface at the sample points. The tangent direction of the curve $Q(\cdot, a_i)$ at $s = s_i$ is equivalent to $\nabla_{a_i}\hat{Q}_i$. Since these tangent directions provide more information about the local shape of the $Q$ surface, it is expected to boost the convergence of $Q$ to its ground truth, which is reflected in Figure 2.

We can also use this example to understand the effects of $w_1$ and $w_2$ on the convergence of $Q$: Figure 2 (c and d) demonstrate that $w_2$, or the weight for $\nabla_{s_i}\hat{Q}_i$, has a more evident effect on the convergence of $Q$. This can be partially explained by the fact that the ground truth $Q$ surface has little variation in the $a$ (action) direction, so knowing the partial derivative $\nabla_{a_i}\hat{Q}_i$ adds very little new information to the fitting problem.

## 4 IMPLEMENTATION

### 4.1 DIFFERENTIABLE SIMULATORS

We test our hybrid algorithm on robot control tasks using a differentiable rigid body simulator, over three different types of settings related to the types of contact experienced:

**Contact-free simulation**  Many classical control examples are contact-free and useful for evaluating and understanding the performance of control algorithms (e.g. Pendulum, Acrobot, *etc.*). The equations of the motion of these problems can be compactly expressed in the form of the standard manipulator equations (Tedrake, 2019), allowing for the simple extraction of system gradients.

**Simulation with impulse-based collision responses**  Due to the significant changes in motion that occur during collisions, rigid body simulations with contacts possess much more complex dynamics than those which are contact-free. In this setting, we employ the Poisson collision model (Moore & Wilhelms, 1988). During collision, an impulse is applied to the rigid body, instantaneously changing the velocity of the rigid body while conserving the momentum of the entire system. For simulating systems with such contacts, we follow (Popović et al., 2000).

**Simulation with constraint-based collision responses** In order to simulate realistic motions for resting and collision contacts among rigid bodies, constraint-based methods formulate contacts as constrained systems, posing them as linear complementary problems (LCP) (Cline & Pai, 2003). Such simulation is accurate but much more complex than previous, simpler soft contact models. Inspired by the work of Amos & Kolter (2017), we implemented a 2D LCP-based rigid body simulator and differentiate through contacts.

## 4.2 THE HYBRID ALGORITHM

Our hybrid control algorithm is implemented based on DDPG in OpenAI Baselines (Dhariwal et al., 2017). Parameter space noise is applied to the actor network for exploration as proposed in (Plappert et al., 2017). Slightly different from the mathematical definition, the critic network in this implementation takes normalized state and action as input, and outputs a normalized $Q$ value in order to achieve task-irrelevant statistical stability. The mean value and standard deviation of state and Q value are empirically estimated from the replay buffer. Actions are scaled by a constant such that they fall in range $[-1, 1]$. Similarly, gradients in line 11 of Algorithm 1 are computed with normalized $Q$, states and actions.

## 5 RESULTS

### 5.1 ENVIRONMENTS

We present seven 2D control tasks implemented in our differentiable rigid body simulator. The tasks are of varying difficulty, including five classic control problems (CartPole, CartPoleSwingUp, Pendulum, MountainCar, and Acrobot), one problem with impulse-based collision response (RollingDie), and one complex control problem with LCP-based contact model (HalfCheetah).

**Classic control environments** Our implementation of CartPole, CartPoleSwingUp, Pendulum, MountainCar, and Acrobot is based on the implementation of OpenAI Gym (Brockman et al., 2016) but with gradient information manually computed in the simulation. We ask readers to refer to (Brockman et al., 2016) for more details.

**Environment with impulse-based collision responses** In order to demonstrate applicability in the context of collisions, we designed a RollingDie task with impulse-based collision responses. The die is thrown from the air to the ground. Its initial state $s_0$ includes position in $x$ and $y$, rotation angle, velocity in $x$ and $y$, and angular velocity. The action $a$ is the torque to the die, i.e. each step, the die can apply an internal torque so as to control its angular velocity (and thus its pose). The reward is defined based on the $L2-$norm of the distance between die center and the target position. Though for most of its trajectory the die is free falling and thus its position cannot be impacted by the torque, the angular velocity and torque do impact the angle with which the die impacts the ground and the resulting bounce. This is a complex control task in which sparse, highly important moments (contact) can chaotically affect the trajectory of the die. Note that the die is allowed to bounce multiple times during the simulation, which increases the complexity of finding an optimized control.

**Environment with LCP-based collision responses** We designed a differentiable HalfCheetah example with LCP-based contacts. The HalfCheetah shares a similar structure as the one in MuJoCo (Todorov et al., 2012) but only has two links on each leg. The reward is defined as $v_x - 0.1\mathbf{u}^T\mathbf{u}$, promoting forward velocity while penalizing highly actuated actions. Note that unlike MuJoCo's soft contact model, the hard contact model we implemented introduces an additional complexity and higher magnitude gradients, making it a more difficult control task.

### 5.2 EVALUATION

We compare our method with three baselines: DDPG, the RL algorithm that we base our method on, MPC with iterative Linear Quadratic Regulartor (iLQR) (Tassa et al., 2014), a state-of-the-art model-based control algorithms employed in previous work on differentiable simulators (Gu et al., 2016), and gradient descent (GD), which employs a neural network actor and Adam (Kingma & Ba,

2014) to directly optimize a final return using the analytical gradients (as in (Hu et al., 2019) and (Degrave et al., 2019)). We repeat each algorithm 16 times on each task with different initializations.

**Initialization**    To ensure a fair comparison between all methods, we seed each algorithm with the same network models (if applicable). Since MPC does not require a neural network but directly solves for state and action variables, we initialize these variables by performing one rollout and collecting the state-action $(s_t, a_t)$ pairs for initialization, using the same initial actor network.

**Metrics**    For methods using a neural network controller (ours, DDPG, and GD), we report the following metrics: *return_mean*, which is the return from one episode generated by running the intermediate actor network every 800 timesteps, and *return_history*, which is the average of the latest 100 *return_mean*. For MPC, as it solves $a_t$ sequentially, *return_mean* is stored whenever $a_t$ is returned. All returns are rescaled linearly so that the average return of a random controller is mapped to 0 and $\max(r) \times T$ is mapped to 1.

**Results and discussions**    Figure 4 and Table 1 summarize the performance and averaged rewards of each method on the seven environments. Overall, our algorithm improves the sampling efficiency of DDPG and often discovers strategies with higher return. Compared with MPC and GD, our method suffers less from local minima, especially for more complex tasks such as HalfCheetah. For all environments, the experiment is terminated when *return_mean* plateaus in GD, when MPC converges, or when the number of timesteps reaches pre-defined maximum number in DDPG and our algorithm. Unlike the other methods, MPC does not perform many sequential simulations, but rather performs one simulation optimized online. In order to perform as fair of a comparison as possible, we plot the predicted cumulative return (based on the previously simulated steps, current planned horizon, and remaining initialized action vector) as it evolves throughout the online planning procedure. As a result, MPC may terminate without using the same number of function evaluations as RL algorithms or showing a sign of plateauing return-timestep curves. We do not compare MPC for the simpler control tasks (Pendulum, MountainCar, CartPole) that are trivial for the algorithm to solve. We also do not compare MPC for the Acrobot, for which the algorithm cannot be applied (discussed later). We discuss the performances of these methods in each environment below.

Immediately obvious from our results is the fact that DDPG and our algorithm are both competitive on all problems presented, regardless of problem difficulty. While MPC dominates on the simplest control tasks, it struggles on the more complicated tasks with hard contacts, and DRL approaches dominate. This underscores our thesis — that DRL's exploration properties make it better suited than model-based approaches for problems with a myriad of poor local minima. More naïve model-based approaches, such as GD, can succeed when they begin very close to a local minimum — as is the case with CartPole — but show slow or no improvement in dynamical environments with nontrivial control schemes. This is especially apparent in problems where the optimal strategy requires robots to make locally suboptimal motions in order to build up momentum to be used to escape local minima *later*. Examples include Pendulum, CartPoleSwingUp, and MountainCar, where the robot must learn to build up momentum through local oscillations before attempting to reach a goal. GD further fails on complex physical control tasks like HalfCheetah, where certain configurations, such as toppling, can be unrecoverable. Finally, we note that although MPC is able to tractably find a good solution for the RollingDie problem, the complex nonlinearities in the contact-heavy dynamics require long planning horizons (100 steps, chosen by running hyperparameter search) in order to find a good trajectory. Thus, although MPC eventually converges to a control sequence with very high reward, it requires abundant computation to converge. DRL-based control approaches are able to find success on all problems, and are especially competitive on those with contact.

Compared with DDPG, our hybrid algorithm universally converges faster or to higher returns. The rolling die example presents a particularly interesting contrast. As the die is randomly initialized, it is more valuable to aim for higher *return_history* rather than *return_mean* due to the large variance in the initial state distribution. It can be seen from Figure 4 that our method managed to reach a higher average *return_history* over 16 runs. Manually visualizing the controller from the best run in our method revealed that it discovered a novel two-bounce strategy for challenging initial poses (Figure 3), while most of the strategies in DDPG typically leveraged one bounce only.

There are a few other reasons why our algorithm may be considered superior to MPC. First, our algorithm is applicable to a wider range of reward structures. While we had planned to demonstrate

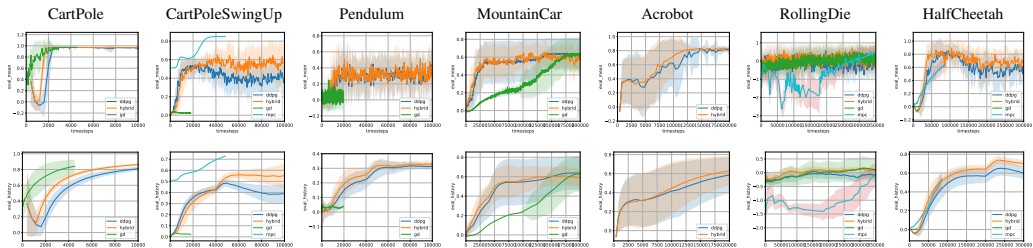

Figure 4: Learning curves of different algorithms (with "Hybrid" being ours) on all 7 tasks, summarized over 16 runs with random initialization for each algorithm. *Return_mean* and *Return_history* are demonstrated in the first and second row respectively. Solid curves represent mean reward over 16 runs, and shaded area shows its standard deviation.

Table 1: The final average return (*return_history*) from baseline algorithms and ours. Best results in each environment are in bold.

|      | CartPole | SwingUp | Pendulum | MountainCar | Acrobot | RollingDie | HalfCheetah |
|------|----------|---------|----------|-------------|---------|------------|-------------|
| DDPG | 0.809    | 0.388   | 0.318    | **0.681**   | 0.586   | -0.136     | 0.599       |
| MPC  | -        | **0.728** | -      | -           | N/A     | -0.077     | 0.545       |
| GD   | 0.842    | 0.023   | 0.039    | 0.628       | N/A     | **0.106**  | 0.003       |
| Ours | **0.862** | 0.552  | **0.327** | 0.587      | **0.630** | 0.084    | **0.701**   |

MPC on another classic control problem, namely the Acrobot, MPC is inapplicable to this robot's reward structure. The Acrobot's rewards penalize it with $-1$ point for every second it has not reached its target pose. MPC requires a differentiable reward, and this reward structure is not. Thus, our Hybrid DDPG algorithm applies to a wider range of problems than MPC. Second, closed-loop network controllers are naturally more robust than MPC. Even as noise is added or initial conditions and tasks change, learned controllers can generalize. While MPC can recover from these scenarios, it requires expensive replanning. In these scenarios, MPC becomes especially unattractive to deploy on physical hardware, where power and computational resource constraints can render MPC inapplicable to realtime applications.

## 6   CONCLUSION

In this paper, we have presented an actor-critic algorithm that uses AE gradients to co-learn critic value and gradient estimation and improve convergence of both actor and critic. Our algorithm leverages differentiable simulation and combines the benefits of model-based methods and DRL. We designed seven 2D control tasks with three different contact scenarios and compared our method with several state-of-the-art baseline algorithms. We demonstrated our method boosts the performance of DDPG and is much less sensitive to local minima than model-based approaches. In the future, it would be interesting to see if our mathematical framework can be applied to improve the effectiveness of value functions used in other DRL algorithms.

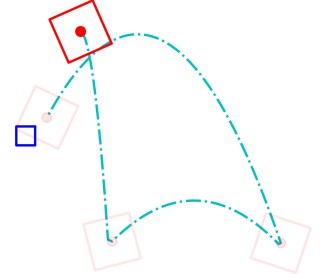

Figure 3: Visualization of the two-bounce strategy discovered by our algorithm. Solid red box: initial die. Dash cyan curve: trajectory of the die. Blue box: the target zone. Light red boxes: states of the die at collisions and about to enter the target.

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

## A  APPENDIX

### A.1  THE CHOICE OF LOSS FUNCTION IN ALGORITHM 1

In gradient-based optimization algorithms, the direction of gradients often plays a more crucial role than its length, especially when adaptive learning rates are applied. Here we experiment with the idea of penalizing the cosine angle between $\nabla Q_\phi$ and $\nabla \hat{Q}$ instead of their L$_2$ distance in Algorithm 1. In particular, we replace line 10 with the following:

$$L = \frac{1}{n}\sum_i \|Q_\phi(s_i, a_i) - \hat{Q}_i\|^2 + w_1(1 - \frac{\nabla_{a_i}Q_\phi \cdot \nabla_{a_i}\hat{Q}_i}{\|\nabla_{a_i}Q_\phi\|\|\nabla_{a_i}\hat{Q}_i\|}) + w_2(1 - \frac{\nabla_{s_i}Q_\phi \cdot \nabla_{s_i}\hat{Q}_i}{\|\nabla_{s_i}Q_\phi\|\|\nabla_{s_i}\hat{Q}_i\|}) \quad (3)$$

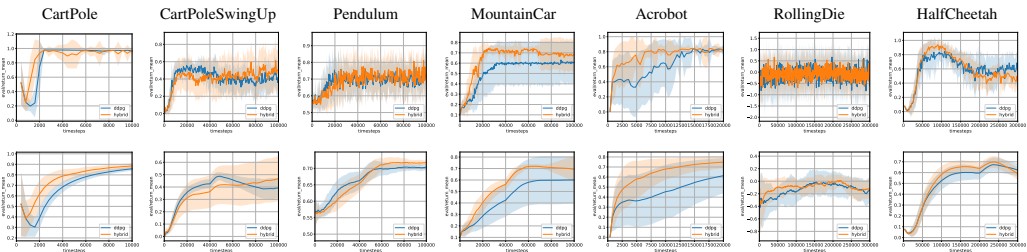

Figure 5: Comparison of the hybrid algorithm with cosine angle regularization with un-regularized DDPG. In all examples, our hybrid algorithm achieves similar or better performance than pure DDPG.

Figure 5 shows the experimental results in our differentiable environments with the new loss function. For each environment, we started with the same parameters as in Figure 4 and decreased $(w_1, w_2) \in \{1, 0.1, 0.01\}$ only if necessary. We ended up using the same parameters as before in MountainCar and Acrobot and switched weights in other environments.

Comparing Figure 4 with Figure 5, we see the cosine angle regularizer is more effective in some examples (particularly MountainCar and Acrobot) than the original L2 norm. The $L_2$ regularizer outperforms the cosine angle for CartPoleSwingUp. The two regularizers perform similarly on the remaining examples.

## A.2 RESULTS ON SOFT ACTOR-CRITIC (SAC)

Apart from DDPG, we have also implemented our method with the original $L_2$-norm loss in the Soft Actor-Critic (SAC) (Haarnoja et al., 2018a) algorithm. Figure 6 reports our experimental results in five examples, among which our method improves the performance of the original SAC in three of them (CartPoleSwingUp, Pendulum, and MountainCar) and performs similarly in the other two. These results have demonstrated that our proposed modification can be generalized to other actor-critic methods other than DDPG.

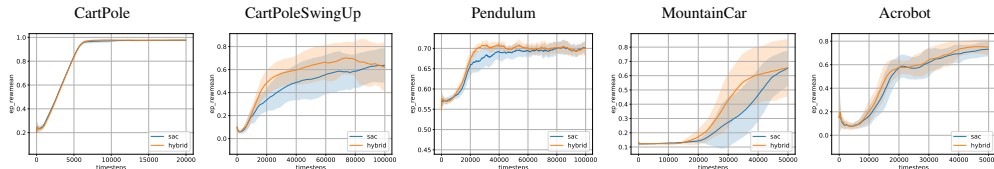

Figure 6: Comparison of the hybrid algorithm with $L_2$ norm regularization with un-regularized SAC. In all examples, our hybrid algorithm performs similar to or better than pure SAC.

