# OpenReview forum: "D3PG: Deep Differentiable Deterministic Policy Gradients"
_ICLR.cc/2020/Conference — Reject_

### Official Review · AnonReviewer3 · 2019-10-24
**Official Blind Review #3**

**Rating:** 3

**Review:**

This paper studies optimal control problems where a physical simulator of the system is available, which outputs the gradient of the dynamics. Using the gradients proposed by the model, the authors propose to add two additional terms in the loss function for critic training in DDPG, where these to terms corresponding to the prediction error of $\nabla_{a} Q(s,a)$ and $\nabla_b Q(s,a)$, respectively. However, my main concern is that the form of gradient given in equation (2) might contains an error.

1. Equation (2). Note that in DDPG, the action is given by a deterministic policy. Thus, we have $a_t = \pi(s_t)$ for all $t\geq 0$. For critic estimation, it seems you are basing on the Bellman equation
$ Q(s,a) = r(s,a) + Q(s', \pi(s'))$, where $s'$ is the next state following $(s,a)$. Then, it seems that Equation (2) is obtained by taking gradient with respect to $(s,a)$. However, I cannot understand what $\nabla_{\pi} Q$ stands for. If it is $\nabla_a Q(s_{i+1}, a_{i+1}) \cdot \nabla_s \pi(s_{i+1}) $, then that makes sense.

2. Based on the experiments, it seems that the proposed method does not always outperform MPC or DDPG, even in a small-scale control problem Mountaincar. Moreover, it seems that the performance is similar to that of the DDPG.

3. Here the model-based gradient in equation (2) is defined by only unroll one-step forward by going from $s_i, a_i$ to $s_{i+1}$. It would be interesting to see how the number of unroll steps affect the algorithm, which is a gradient version of TD($\lambda$).

4. Missing reference: Differential Temporal Difference Learning https://arxiv.org/abs/1812.11137

**Experience Assessment:**

I have published in this field for several years.

**Review Assessment: Checking Correctness Of Derivations And Theory:**

I carefully checked the derivations and theory.

**Review Assessment: Checking Correctness Of Experiments:**

I carefully checked the experiments.

**Review Assessment: Thoroughness In Paper Reading:**

I read the paper thoroughly.

---

> ### Author Response · Authors · 2019-11-06
> **Clarifications on the correctness of Equation (2)**
>
> Thank you for your constructive review!
>
> We really appreciate your comments and are happy to discuss them during this rebuttal period. But for now, we just want to make a quick clarification on Equation (2) and justify our gradient computation:
>
> You are correct that we base our computation on the Bellman equation. To be precise, we use $\hat{Q}(s_i,a_i)=r(s_i,a_i,s_{i+1})+\gamma Q'(s_{i+1},\pi'(s_{i+1}))$. This is line 9 in Algorithm 1 in our paper, and it is also consistent with line 12 in Algorithm 1 in the original DDPG paper.
>
> For brevity, in Equation (2) we use the neural network names to refer to its output value. So $\pi'$ in Equation (2) stands for $\pi'(s_{i+1})$ and $\nabla_{\pi'}Q'$ in Equation (2) stands for:
> $$
> \nabla_aQ'(s,a)|_{s=s_{i+1},a=\pi'(s_{i+1})}
> $$
> You can also check the correctness of Equation (2) by comparing it to the computation graph in Figure 1, where the upper right $\mu$ stands for $\pi'(s_{i+1})$. In Figure 1, $\nabla_{\pi'}Q'$ corresponds to the gradient back-propagated along the arrow $\mu\rightarrow Q'$ ("if we change $\mu$, how much will $Q'$ change?"). Similarly, the term $\nabla_{s_{i+1}}\pi'$ after $\nabla_{\pi'}Q'$ in Equation (2) corresponds to the arrow $s_{i+1}\rightarrow\mu$ ("if we change $s_{i+1}$, how much will $\mu$ change?").
>
> Putting them together, the product $\nabla_{\pi'}Q'\cdot \nabla_{s_{i+1}}\pi'$ in Equation (2) back-propagates the gradient along the path $s_{i+1}\rightarrow\mu\rightarrow Q'$ in Figure 1. Similarly, the term $\nabla_{s_{i+1}}Q'$ in Equation (2) corresponds to the arrow $s_{i+1}\rightarrow Q'$ in Figure 1, and the sum $(\nabla_{s_{i+1}}Q'+\nabla_{\pi'}Q'\cdot \nabla_{s_{i+1}}\pi')$ computes the total derivative of $Q'$ with respect to $s_{i+1}$. The other terms in Equation (2) can be verified in the same way.
>
> We hope this explanation can clear your concern with the correctness of Equation (2). We will revise the manuscript to clarify the notations.

---

> ### Author Response · Authors · 2019-11-14
> **Using a gradient version of TD($\lambda$)**
>
> We agree that using the gradients from TD($\lambda$) in Equation (2) is an interesting direction to explore. However, unrolling more steps to estimate $\hat{Q}_i$ requires more on-policy samples: for example, unrolling one more step in line 9 of algorithm 1 would require access to $s_{i+2}$ computed by simulating the robot from $(s_{i+1}, \pi’(s_{i+1}))$. These new samples are not directly available from the off-policy replay buffer in DDPG and have to be regenerated on the fly, which hurts the sampling efficiency of the algorithm.
>
> We did think about applying the same technique to on-policy RL algorithms and have implemented the TD($\lambda$) version of equation (2) in PPO. Our preliminary results showed that it did not improve the performance of PPO even after hyperparameter tuning. We suspect the reason is that Equation (2) assumes the policy is deterministic in nature while PPO uses stochastic policies. Still, we think it is possible that a proper combination of TD($\lambda$) gradients and RL baseline algorithms could lead to an improvement in performance.

---

### Official Review · AnonReviewer1 · 2019-10-25
**Official Blind Review #1**

**Rating:** 3

**Review:**

This paper shows how the derivatives from a differentiable
environment can be used to improve the convergence rate of
the actor and critic in DDPG.
This is useful information to use as most physics simulators
have derivative information available that would be useful
to leverage when training models.
The empirical results show that their method of adding
this information (D3PG) slightly improves DDPG's
performance in the tasks they consider.
As the contribution of this work is empirical is nature,
I think a very promising future direction fo work is to
add derivative information to and evaluate similar
variants of some of the newer actor-critic methods
such as TD3 and SAC.

I have two minor questions:
1) Figure 2(a) shows the convenrgence of regularizing states,
   actions, and both states and actions and the text
   describing the figure states that this is
   "expected to boost the convergence of Q."
   However the figure shows that regularizing both states and
   actions results in a slower convergence than doing
   them separately. Why is this?
2) How should I interpret the visualization of the
   learned Q surface in Figure 2(f) in comparison to
   the true Q function in Figure 2(g)?
   It does not look like a good approximation.

**Experience Assessment:**

I have published one or two papers in this area.

**Review Assessment: Checking Correctness Of Derivations And Theory:**

I assessed the sensibility of the derivations and theory.

**Review Assessment: Checking Correctness Of Experiments:**

I assessed the sensibility of the experiments.

**Review Assessment: Thoroughness In Paper Reading:**

I read the paper at least twice and used my best judgement in assessing the paper.

---

> ### Author Response · Authors · 2019-11-15
> **Experimenting with SAC and interpreting results from the motivating example**
>
> Thank you for your constructive review!
>
> == New actor-critic methods ==
> We agree that TD3 and SAC are good candidates to try besides DDPG. We have implemented a variant of SAC and reported the results in five examples in Section A.2 of the updated manuscript. Our experiments showed that the proposed method helped improve the performance of the original SAC in three examples and obtained similar performance in the other two.
>
> == Q function approximation ==
> The Q network does not fit the ground-truth closely because 1) The RL algorithm only explored and used a very small part of the whole domain of ($s$, $a$) for fitting. Specifically, since samples were extracted from perturbing $\pi$, most of them clustered around the curve $(s, \pi(s))$; 2) Due to the design of this problem, regions far away from the initial ($s=-0.5$) and final ($s=0$) positions of the mass point are rarely visited during training and not needed in the final solution. Due to these two reasons, the Q network attempted to fit the ground-truth Q well only in the banded region between $s=-0.5$ and $s=0$, and it can be observed that adding weighted loss on gradient differences helped the Q network converge to the ground-truth in this banded area faster.
>
> == Slower convergence when both weights are available ==
> Due to the empirical nature of our method, we are not able to justify this phenomenon on a theoretical basis. We suspect it might be related to the fact that the Q function in this example has different sensitivity to its two inputs $s$ and $a$. In particular, if we slice the ground-truth Q surface at a given $s$, the resulting $Q-a$ curve is very flat, so more gradient information about $\partial Q/\partial a$ might be unnecessary and not helpful for fitting it well.

---

> > ### Comment · AnonReviewer1 · 2019-11-15
> > **Thanks for the response and additional experiments!**
> >
> > Thanks for the clarifications and additional experiments using this with SAC as well, which is exactly what I suggested in my original review. I've also read through the other reviews and responses in this thread. If there were an option to update my score to "neutral" I would increase it to this, as the gradient information this paper adds is interesting and relevant to the community, but the empirical results in the current form are still difficult to distinguish from the baselines.

---

### Official Review · AnonReviewer2 · 2019-10-30
**Official Blind Review #2**

**Rating:** 6

**Review:**

==Summary==

DDPG is a popular RL method for continuous control problems. It is more widely applicable than traditional model-based approaches like MPC, since it doesn't require differentiable models of the dynamics. However, in many environments, dynamics are differentiable. This paper proposes a method for extending DDPG to exploit simulator gradients. In particular, the Bellman error objective (which is defined in terms of critic values) used for training the critic is augmented with additional terms defined in terms of gradients of the critic. This leads to faster convergence in practice on a range of benchmarks.

==Overall Assessment==

I recommend acceptance. The paper's contribution is well-motivated, works reasonably well, and is relatively easy to implement.

==Comments==

It would be good to add an argument explaining to readers that accurately estimating Q using Q_\phi does not mean that the gradients of Q_\phi will be good approximations of the true gradients of Q. I found Fig 1 of arxiv.org/pdf/1705.07107.pdf informative.

Can you justify the choice of euclidean norm in line 10? In terms of the critic helping teach the actor, the direction of the gradient may be more important than the norm. What if you used cosine sim?

You argue that DRL is better than MPC because DRL explores better. Could you use the simulator gradients somehow to improve exploration?


**Experience Assessment:**

I have read many papers in this area.

**Review Assessment: Checking Correctness Of Derivations And Theory:**

I assessed the sensibility of the derivations and theory.

**Review Assessment: Checking Correctness Of Experiments:**

I assessed the sensibility of the experiments.

**Review Assessment: Thoroughness In Paper Reading:**

I read the paper at least twice and used my best judgement in assessing the paper.

---

> ### Author Response · Authors · 2019-11-14
> **Experimental results on cosine similarity and thoughts on using gradients for exploration**
>
> Thank you for your constructive feedback!
>
> Thank you for sharing Fig. 1 in "Gradient Estimators for Implicit Models". We agree that such an example would highlight the shortcomings of estimating only the Q function and the benefit of training with simulation gradients whenever they are available. We will consider including this argument in a stronger motivating example and adding a similar figure in the manuscript.
>
> We agree comparing other norms is an interesting idea and norms like L1 and cosine would both be interesting to try.  For now, we have tested the cosine norm on all of our examples.  For some of our examples (the Acrobot and MountainCar), we found that the cosine norm dominates the L2 norm.  For the CartPoleSwingUp, the L2 norm still dominates.  For the remaining problems, both norms work approximately equally well. We will include these quantitative results in the revised manuscript. We stress that in all cases, both regularization variants achieve performance similar to or better than pure DDPG.  It is difficult to give a precise theoretical reason as to why one norm outperforms the other for certain problems, however, we can gladly report extensive empirical findings in a final version of the manuscript.
>
> Simulator gradients are unfortunately difficult to use to directly improve exploration since they always point in a greedy direction.  In the classic exploration/exploitation tradeoff, the gradient provides exploitation.  It is possible that one could devise an algorithm that may improve exploration by sampling updates which deviate from the deterministic gradient (e.g. a gradient-based variant of https://arxiv.org/pdf/1706.01905.pdf).  However, such an algorithm could be tried with or without gradient fitting.  It is possible that gradient-fitting would improve the efficacy of such a technique, but this is all introducing a new, potentially complex algorithm, worthy of its own manuscript and study.

---

### Author Response · Authors · 2019-11-15
**Paper update**

We thank all reviewers again for their feedback. We have uploaded a new manuscript based on the review and we summarized our updates below:

1. We added clarification to the notations in Equation (2) and updated the computational graph (Figure 1) to better explain the computation of our gradients.
2. We reported experimental results in the Appendix (Section A.1) about using cosine similarity instead of L2 norm in all examples. These experiments showed that our method is not sensitive to the choice between these two norms, and both norms are viable options for our examples.
3. We implemented our proposed method in another actor-critic method (SAC) and reported the experimental results in five of our examples. We use these experiments to demonstrate that it is possible to apply our method to other actor-critic methods besides DDPG.

We hope these updates can help articulate the benefits of incorporating gradient information in RL training whenever a differentiable simulator is available. Please feel free to leave more comments and thank you again for your review!

---

### Decision · Program_Chairs · 2019-12-19

**Decision:**

Reject

**Comment:**

This paper proposes a hybrid RL algorithm that uses model based gradients from a differentiable simulator to accelerate learning of a model-free policy.  While the method seems sound, the reviewers raised concerns about the experimental evaluation, particularly lack of comparisons to prior works, and that the experiments do not show a clear improvement over the base algorithms that do not make use of the differentiable dynamics. I recommend rejecting this paper, since it is not obvious from the results that the increased complexity of the method can be justified by a better performance, particularly since the method requires access to a simulator, which is not available for real world experiments where sample complexity matters more.